# Differences in the Composition of Organic Aerosols between Winter and Summer in Beijing: a Study by Direct Infusion Ultrahigh Resolution Mass Spectrometry

Sarah S. Steimer[1,2,a], Daniel J. Patton[1], Tuan V. Vu[3], Marios Panagi[4,5], Paul S. Monks[6], Roy M. Harrison[3,7], Zoë L. Fleming[4,b], Zongbo Shi[3] and Markus Kalberer[1,2]

[1]Department of Chemistry, University of Cambridge, Cambridge, CB2 1EW, UK
[2]Department of Environmental Sciences, University of Basel, 4056 Basel, Switzerland
[3] Division of Environmental Health & Risk Management, School of Geography, Earth & Environmental Sciences, University of Birmingham, Birmingham, B1 52TT, UK
[4]National Centre for Atmospheric Science (NCAS), Department of Chemistry, University of Leicester, Leicester, UK
[5]Department of Physics and Astronomy, Earth Observation Science Group, University of Leicester, Leicester, UK
[6]Department of Chemistry, University of Leicester, Leicester, UK
[7]Department of Environmental Sciences/Center of Excellence in Environmental Studies, King Abdulaziz University, P.O. Box 80203, Jeddah, Saudi Arabia
[a]now at: Department of Environmental Science, Stockholm University, 106 91 Stockholm, Sweden
[b]now at: Center for Climate and Resilience Research (CR2), Departamento de Geofísica, Universidad de Chile, Santiago, Chile

*Correspondence to*: Sarah S. Steimer (Sarah.Steimer@aces.su.se)

**Abstract.** This study investigates the chemical composition of $PM_{2.5}$ collected at a central location in Beijing, China, during winter 2016 and summer 2017. The samples were characterised using direct infusion negative nano-electrospray ionisation ultrahigh resolution mass spectrometry to elucidate the composition and the potential primary and secondary sources of the organic fraction. The samples from the two seasons were compared with those from a road-tunnel site and an urban background site in Birmingham, UK, analysed in the course of an earlier study using the same method. There were strong differences in aerosol particle composition between the seasons, particularly regarding (poly-)aromatic compounds, which were strongly enhanced in winter, likely due to increased fossil fuel and biomass burning for heating. In addition to the seasonal differences, compositional differences between high and low pollution conditions were observed, with the contribution of sulfur-containing organic compounds strongly enhanced under high pollution conditions. There was a correlation of the number of sulphur-containing molecular formulae with the concentration of particulate sulfate, consistent with a particle-phase formation process.

# 1 Introduction

Ambient air pollution is of growing concern regarding its negative effect on public health, especially in low-and middle-income countries (Cohen et al., 2017; Hoek et al., 2013; Lelieveld et al., 2015). One of these countries is China, where rapid development has led to air pollution becoming a major environmental issue (Guan et al., 2016). Areas of strong industrial activity and rapid urbanisation are particularly impacted (Chan and Yao, 2008; Hu et al., 2014). Beijing, China's capital, has faced severe issues with air pollution in recent decades, with a particular impact of particulate pollution (Zhang et al., 2016). As a response to this problem, an international research collaboration, the Atmospheric Pollution and Human Health in a Chinese Megacity (APHH Beijing) project, was launched in an effort to understand the emissions, processes and health effects of air pollution in Beijing (Shi et al., 2019). This improved scientific understanding will then feed into the development of efficient mitigation measures to improve air quality and reduce health impacts. As a central part of the project, two month-long coordinated field campaigns were conducted at two sites, central Beijing and rural Pinggu, in November – December 2016 and May- June 2017.

Fine ambient particulate matter (PM) is the major air pollutant (Dockery et al., 1993; Pope III et al., 2002), estimated to cause about 3 million deaths per year (World Health Organization, 2016). About 20-90% of the fine particle mass is organic (Jimenez et al., 2009; Kanakidou et al., 2005). Organic material in the atmosphere is highly complex, consisting of thousands of different compounds (Goldstein and Galbally, 2007). Ultra-high- resolution MS (UHRMS) analysers such as the Fourier transform ion cyclotron (FTICR) or the Orbitrap have a high mass resolving power and high mass accuracy. This allows for separation of 100s to 1000s of compounds within a relatively small mass range, usually 100-500Da, even without further chromatographic separation (Nozière et al., 2015). Additionally, it is possible to assign molecular formulae to unknown compounds. Direct infusion UHRMS has proven to be highly successful in assessing the chemical properties of ambient aerosols from a large variety of sampling locations, ranging from remote (e.g. Dzepina et al., 2015; Kourtchev et al., 2013) to rural (e.g. Schmitt-Kopplin et al., 2010; Wozniak et al., 2008) and urban (e.g. Giorio et al., 2019; Tao et al., 2014; Tong et al., 2016). Since it is possible to assign elemental compositions to unknown compounds, compounds containing heteroatoms such as sulfur and nitrogen can be unambiguously detected. This has for example been used for the study of organosulfates (Lin et al., 2012; Schmitt-Kopplin et al., 2010; Tao et al., 2014).

In this paper we investigate the chemical composition of the polar organic fraction of $PM_{2.5}$ collected in central Beijing during the APHH-Beijing campaign. While the average annual contribution of organic matter (OM) to $PM_{2.5}$ Beijing has decreased since the 2000's, it remains a major contributor (>20%) (Lang et al., 2017). The contribution of organic carbon to $PM_{2.5}$ is usually highest during winter (He et al., 2001; Lin et al., 2009; Wang et al., 2015). There are numerous studies investigating the sources of PM2.5 in Beijing. Biomass burning, dust, coal combustion, vehicle emissions, cooking, and the secondary products have all been identified as important sources of $PM_{2.5}$ in Beijing (Lv et al., 2016), with varying importance for the different seasons (Yu et al., 2013; Zhang et al., 2013). Coal combustion for residential heating was often found to be the

dominant source of PM$_{2.5}$ during winter (Song et al., 2007; Zhang et al., 2017b), though some recent studies highlight traffic emissions as the winter main source (Gao et al., 2018; Zíková et al., 2016).

In our study, we are focussing on the differences in aerosol composition between summer and winter and the influence of high vs. low pollution conditions. The Beijing data is also compared with data from Birmingham, UK to investigate differences in composition in comparison with European urban background samples as well as with samples which are strongly influenced by traffic emissions.

## 2 Materials and Method

### 2.1 Particle sampling and sample preparation

As part of the APHH-Beijing campaign (Shi et al., 2019), a total of 67 PM$_{2.5}$ aerosol samples were collected over 23 h each on quartz microfiber filters (20.3 x 25.4 cm, Whatman™, GE, USA) at the Institute of Atmospheric Physics (IAP), Chinese Academy of Sciences in Beijing, China, using a high volume sampler (HVS) (TE-6070VFC, Tisch Environmental, USA). This field site is an urban site located in a residential area between the fourth and third North ring roads of Beijing. In addition to the samples, several procedural blank filters were collected. After sampling, all filters were stored at <-22°C till analysis. Of these filters, 33 were sampled during winter from 09 November to 11 December 2016 and 34 were sampled during summer from 22 May to 24 June 2017. For the study presented here, we selected the five filters with the highest mass loading and the five filters with the lowest mass loading for winter and summer each. All of the filters with high mass loadings in winter were impacted by haze events, while this was only the case for two of the five filters with high mass loadings in summer (Shi et al., 2019). An overview of the selected filters and their respective mass loadings is given in Tab. S1. From these 20 filters, filter punches (1.8 or 3.5 cm$^2$) were extracted three times in 5 ml methanol (Optima® LC/MS grade) by sonication. The samples were cooled in an ice water bath during the extraction process to prevent heating of the solvent. The combined extracts from the three rounds of sonication were first filtered using 0.45 µm pore size filters (SUPELCO Iso-Disc™ Filters PTFE-4-4, 4 mm×0.45 µm). After gentle evaporation under nitrogen (N$_2$) to about 1-2 ml, the extracts were filtered a second time using 0.2 µm pore size filters (SUPELCO Iso-Disc™ Filters PTFE-4-2, 4 mm×0.2 µm) and evaporated further under N$_2$ to volumes between 120 µl and to 1.8 ml. The volumes were adjusted so that the concentration of total particle mass in the extracts is approximately the same (~1.0 µg·µl$^{-1}$ assuming complete solvation). In addition, one of the procedural blank filters was extracted using the same method as described above for each of the two seasons. The extracts were stored in the freezer for 1-2 days until mass spectrometry analysis. For the Birmingham samples, five 24 h HVS quartz fibre filters were collected form an urban background site and a road tunnel each and processed in a similar way as described above, although with only one filtration step. A more detailed description of these samples and their collection, analysis and data processing can be found in Tong et al. (2016).

## 2.2 Mass spectrometry measurement and data processing

All mass spectrometry measurements were performed in direct infusion, negative ionisation mode using an ultrahigh resolution LTQ Orbitrap Velos Mass Spectrometer (Thermo Fisher, Bremen, Germany) with a TriVersa NanoMate® chip-based electrospray ionisation source (Advion Biosciences, Ithaca NY, USA). The source parameters were set to an injection volume of 5.0 µl, an ionisation voltage of −1.4 kV and a back pressure of 0.7 psi. The capillary temperature was 200°C. The mass spectrometer was routinely calibrated using Pierce LTQ Velos ESI Positive Ion Calibration Solution and a Pierce ESI Negative Ion Calibration Solution (Thermo Scientific, Waltham, MA, USA). The mass accuracy of the instrument was below 1.5 ppm, which was regularly checked before the analysis. Mass spectra were collected in full scan mode over two different mass ranges: $m/z$ 100-650 and $m/z$ 150-900, with a resolution of 100 000 at $m/z$ 400. For each sample and blank, three replicate measurements of 1 min each were carried out for the two different mass ranges.

Initial assignments of molecular composition were made using Xcalibur 2.2 software (Thermo Scientific). The following constraints were applied to both sample and blank mass spectra: maximum number of molecular formula assignments per peak in the mass spectrum: 40, mass tolerance: ±5 ppm; the molecular formula were assumed to contain only the following elements with the given number of atoms: $1 \leq {}^{12}C \leq 100$, $0 \leq {}^{13}C \leq 1$, $1 \leq {}^{1}H \leq 200$, $0 \leq {}^{16}O \leq 50$, $0 \leq {}^{14}N \leq 5$, $0 \leq {}^{32}S \leq 2$, $0 \leq {}^{34}S \leq 1$. The three repeat measurements of the blank filters for both high and low mass range were manually merged to yield four final blank files: low mass range winter, high mass range winter, low mass range summer and high mass range summer. Each of these merged blank files contains all masses from the three repeat measurements as separate data points. Data filtering was performed using Mathematica 11.2 (Wolfram Research Inc., UK) with a code package developed in-house (Zielinski et al., 2018). In the first instance, all ions below the noise level, which was estimated based on fitting a normal distribution to a histogram of intensities, were removed from the spectrum. As a second step, peaks in the sample that have a corresponding match in the blank with an intensity above a minimum sample-to-blank ratio of 10 were removed (blank subtraction). Based on the mass drift of 12-15 reference compounds present in the sample, the maximum acceptable mass drift was set to the highest/lowest reference mass drift +/-0.5 ppm. All assignments based on mass drifts outside this range were discarded. Several additional rules were employed to remove chemically non-meaningful assignments: all molecular formulae where O/C ≥ 2.0, 0.3 ≤ H/C ≥ 2.5, N/C ≥ 1.3, S/C ≥ 0.8 were eliminated with the aim to remove compounds that are not likely to be observed in nature, as well as assignments without carbon, hydrogen or oxygen. Neutral formulae that had either a noninteger or a negative value of the double bond equivalent (DBE) were also removed from the list of possible molecules. Double bond equivalents were calculated using the following equation (McLafferty and Tureček, 1993):

$$DBE = x - \frac{1}{2}y + \frac{1}{2}z + 1 \qquad (1)$$

with $I_y II_n III_z IV_x$, where I=monovalent elements; II=bivalent elements; III=trivalent elements and IV=tetravalent elements. Sulfur was assumed to be bivalent and nitrogen trivalent for this calculation. Assignments which fail the nitrogen rule (McLafferty and Tureček, 1993) were similarly removed. Elemental formulae containing ${}^{13}C$ or ${}^{34}S$ were checked for the

presence of their [12]C or [32]S counterparts respectively. If there was no peak with a matching composition containing only the lighter isotope or if the intensity ratio of heavier-to-lighter isotope was greater than the natural isotopic abundance, the formula with the next larger mass error was used instead. The three repeated measurements for each sample and mass range were then combined into one file, keeping only ions present in all three replicates. The two mass ranges were then merged, resulting in one combined mass spectrum per sample. As a final step, the five samples taken for each of the four measurement conditions (winter high (WH), winter low (WL), summer high (SH) and summer low (SL)) were combined into a single mass spectrum for each of the four conditions. These mass spectra contain only ions present in all five samples analysed for the respective atmospheric condition, thus representing a typical chemical composition for WH, WL, SH and SL, respectively. With the exception of section 3.3, the following discussion will only compare these four common ion spectra only and not the 20 individual samples.

## 2.3 Ion chromatography analysis

A cut piece of 3 cm$^3$ of theses filters and 10 mL deionized water (18Ω) were added into a 15 mL polypropylene centrifuge tube. They were then sonicated using an ultrasonic bath with ice water for 1 hour at a controlled temperature (<20°C). Subsequently, the extract solution was shaken on a mechanical shaker for three hours at approximately 60 cycles per minute. Water anion soluble ions ($SO_4^{2-}$, $NO_3^-$) in the filtered extract solution were analyzed by ion chromatography (Dionex model ICS-1100).

## 2.4 Back trajectory modelling

The Numerical Atmospheric Modelling Environment (NAME, UK Met Office) (Jones et al., 2007) was used to track the pathways of air masses arriving in Beijing. A large number of hypothetical inert particles are released and their pathways are tracked backwards in time using meteorological fields from the UK Met Office's Unified Model with a horizontal grid resolution of 0.23° longitude by 0.16° latitude and 59 vertical levels up to an approximate height of 30 km (Brown et al., 2012). For this study, we modelled 3 day backward footprints with release periods  that were the same time as the measurements (i.e 22/23 h). The output has a resolution of 0.25° x 0.25°, and represents the hypothetical inert particles passing through the surface layer (here defined as: 0–100 m above ground) during their travel to the IAP meteorological tower at Beijing.  The residence time the air masses spent over a specific location (Fig. 3) or region (Fig. S2) during the four measurement conditions (WH, WL, SH, SL) was calculated by producing a summed plot of the NAME footprints from the five individual samples analysed for each measurement condition. More in-depth information about the origin of different air masses in Beijing, including time periods not covered by the APHH campaign, can be found in Panagi et al. (2020). They investigated the different origins of the air masses during a four year period and how this is correlated with CO levels and CO transportation to Beijing and calculated the residence times in air masses from four quadrants around the IAP tower in Beijing, finding many more north westerly air masses in winter and south in summer.

## 3 Results and Discussion

### 3.1 Compositional Overview

The common ion mass spectra for the four atmospheric conditions are shown in Fig. 1, indicating that the vast majority of compounds above the detection limit are below m/z 500. While in winter almost no peaks are present with m/z >450, a significant number of peaks up to m/z 500 are detected in the summer samples. The overall number of assigned formulae per sample ranged from 918 in the SL sample to 1586 in the WH sample. This is a lower limit on the total number of ionised compounds in the samples, as the technique cannot distinguish between structural isomers. Figure 2 shows the relative number contribution of the four different compound groups (CHO, CHON, CHOS, CHONS) to the total number of assigned formulae. ESI, like all ionisation methods for mass spectrometry, is not equally efficient in ionising different compounds and the ionisation efficiency for one compound will differ between positive and negative mode. ESI works best for polar molecules, which is why pure hydrocarbons are usually not detected efficiently. A trend that can be seen in the detected formulae is the presence of significantly more CHON formulae in summer compared to winter. The opposite trend was observed in Shanghai by Wang et al. (2017a), who found higher numbers and relative contribution of CHON in winter. Further information would be needed to explain this discrepancy. Another seasonal difference is that significantly more sulphur-containing formulae (CHOS and CHONS) were observed for high pollution conditions in both winter and summer. Overall, the highest percentage of sulphur-containing formulae was found under high pollution conditions in winter. Figure S1 shows the overlap of assigned formulae, divided into CHO, CHON, CHOS and CHONS, between the different samples. From this graph, it can be seen that the low mass loading samples (i.e. WL and SL) contain hardly any unique CHOS and CONS formulae above our detection limit. These results regarding the sulfur-containing compounds are in good agreement with Jiang et al. (2016) who compared wintertime Beijing samples collected under hazy and clean conditions. A more detailed discussion of the sulphur and nitrogen containing compounds and their sources can be found in section 3.3.

The strongly varying particle composition in winter and summer and during high and low pollution conditions is reflected by changes of the source regions during these four conditions. Back trajectories show that during strongly polluted days (WH, SH), air masses originate from south of Beijing, which is widely industrialised (Fig. 3). In contrast, days with low pollution conditions in both seasons are characterised by more varied air mass histories. Especially during the collection of the WL samples, the air masses originated predominantly from the northwest (Fig. S2), a region including a less developed part of China as well as the only sparsely populated country Mongolia.

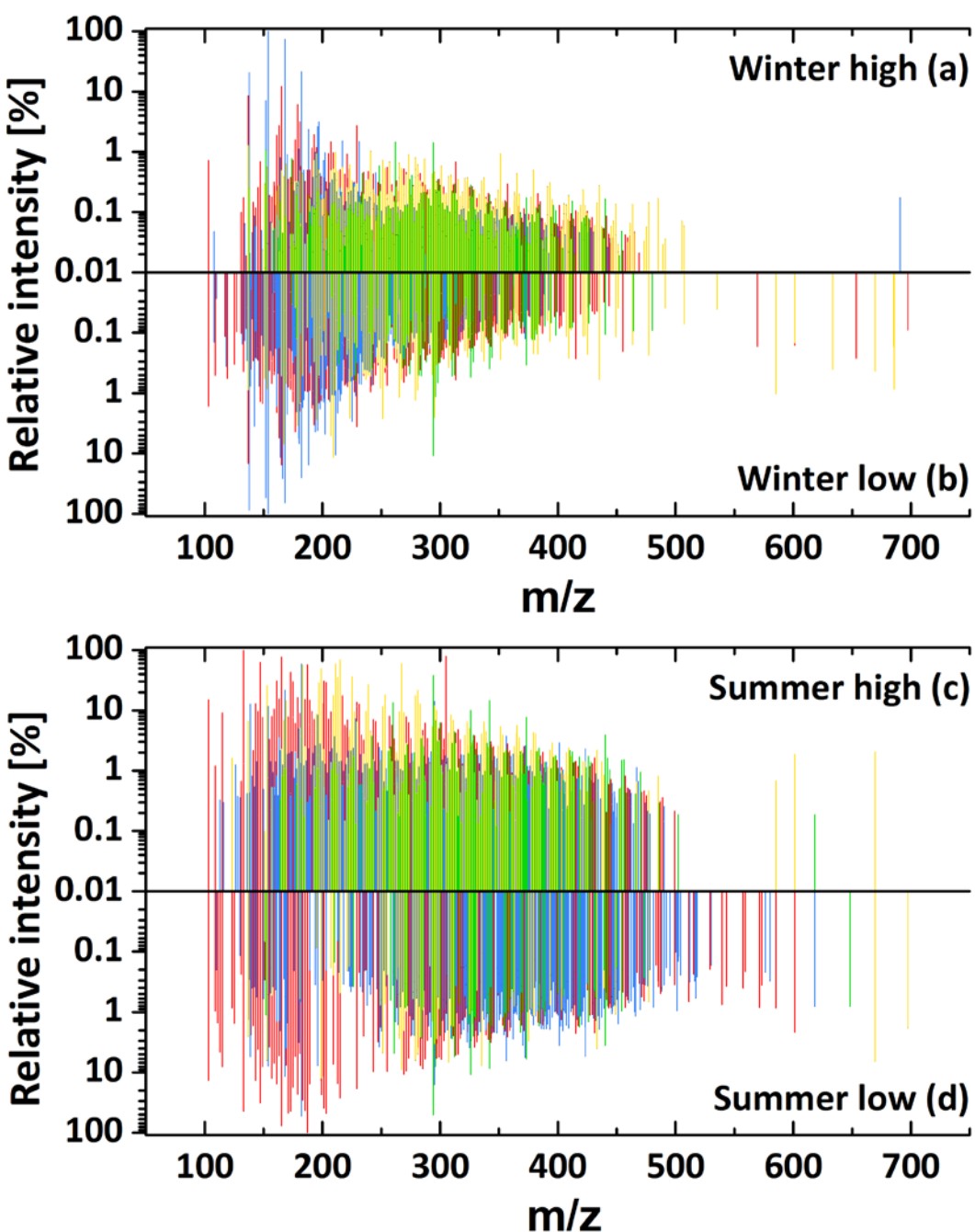

**Figure 1: Negative ionisation mass spectra for the four composite samples WH (a), WL (b) SH (c) and SL (d). Colour-coding differentiated formulae with differing molecular compositions: CHO (red), CHON (blue), CHOS (yellow) and CHONS (green).**

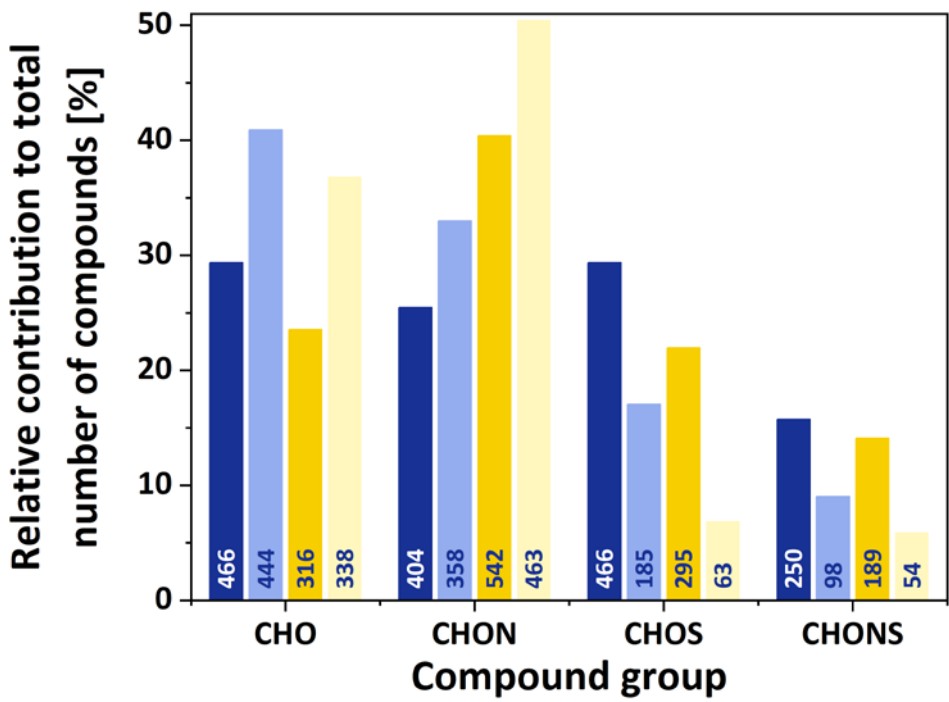

**Figure 2: Bar charts showing the relative contribution of different compound groups (CHO, CHON, CHOS, CHONS) to the total number of formulae for each of the composite samples: WH (dark blue), WL (light blue), SH (dark yellow) and SL (light yellow). The numbers on the bars indicate the absolute number of formulae detected for the respective sample and compound class.**

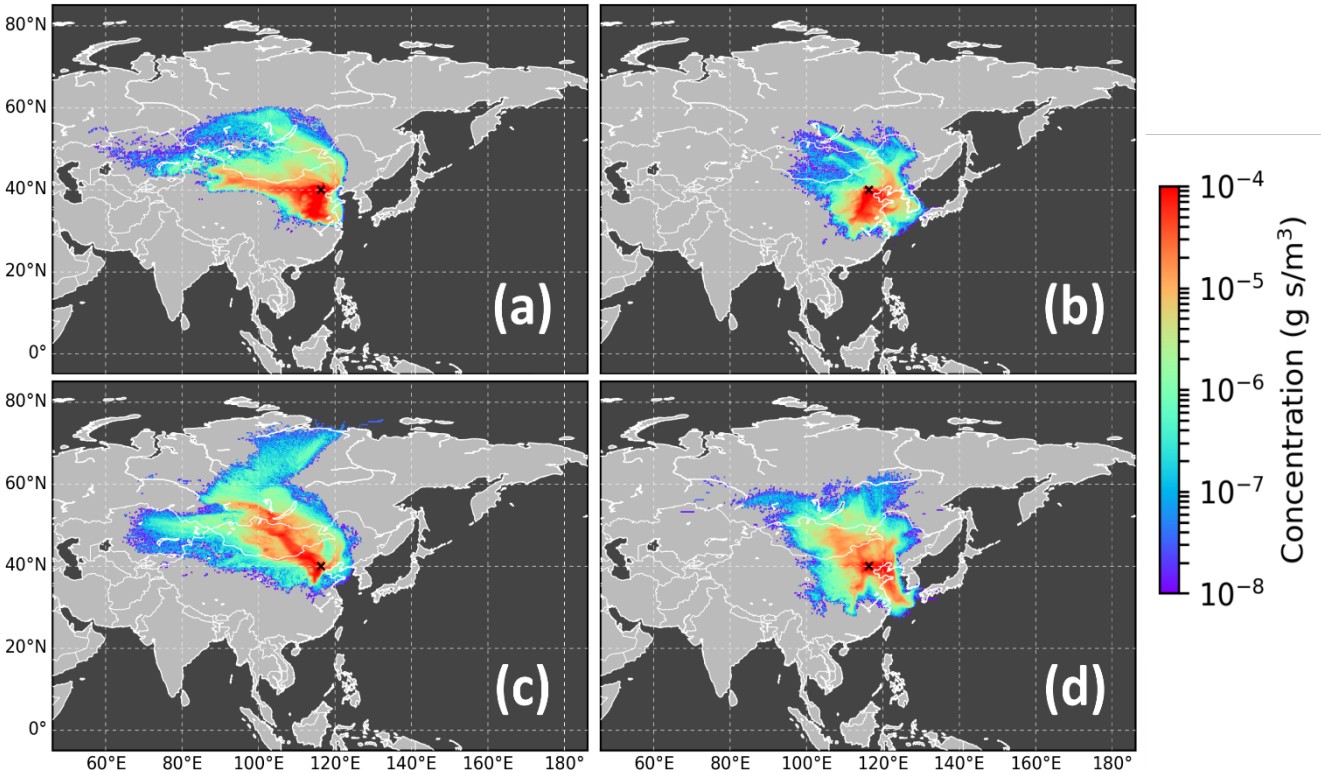

**Figure 3: Sum of the 72 hour back trajectories for a) winter, high pollution days (WH), b) summer, high pollution days (SH), c) winter, low pollution days (WL) and d) summer, low pollution days (SL). The colors denote the relative (on a logarithmic scale) residence time of the air masses in each 0.25 ° x 0.25 ° grid box (up to 100m from the surface) during the last 72 hours before arriving at the monitoring station (the model calculates the concentration of theoretical air mass particles integrated over time per volume). The black cross denotes the sampling location.**

Average oxygen to carbon (O/C) and hydrogen to carbon (H/C) ratios for each sample set were calculated by dividing the total number of oxygen respective hydrogen atoms in all formulae in the data set by the total number of carbon atoms. The calculated ratios are shown in Table 1. The average H/C ratios are lower in winter and in the Birmingham tunnel (BT) sample. This points towards the presence of more aromatic formulae, while the summer and the Birmingham background (BB) sample are more aliphatic. Aromatic compounds are predominantly produced from anthropogenic sources such as traffic, industry and heating (Baek et al., 1991; Hamilton and Lewis, 2003) whereas aliphatic compounds can be of both anthropogenic and biogenic origins, the latter of which are more prevalent in the summer (Gelencsér et al., 2007; Hu et al., 2017; Kleindienst et al., 2007). One type of source that will be present in both seasons and which contributes compounds of both high and low H/C to particulate matter is vehicle emissions, as these are usually a mix of low carbon number (<24) PAHs and single-ring aromatics with low H/C and alkenes and cyclic, branched and straight-chain alkanes with high H/C (Gentner et al., 2012, 2017; Huang et al., 2015; May et al., 2014; Worton et al., 2014). The SL H/C ratio is particularly high which may be due to a larger proportion of primary biogenic organic aerosol components from plant sources with a high H/C, such as plant waxes, and a smaller influence of

industrial sources or vehicle emissions which is more pronounced in the high pollution sample. Increased contribution of biogenic plant waxes to PM$_{2.5}$ during summer in Beijing has been observed previously (Feng et al., 2005).

While the overall H/C ratio shows whether the sample is more aromatic or aliphatic, the O/C ratio gives an indication of how strongly oxidised a sample is. Table 1 shows that there are more highly oxidised formulae in the summer and the BB sample

sets which could be due to higher levels of photochemistry in the summer in Beijing, and at the Birmingham background site. The O/C ratio is much lower in the winter and in the Birmingham tunnel sample. In Beijing, this could be due to reduced photochemistry in the winter months in Beijing, so that the particles are sampled before they can undergo atmospheric ageing processes, for example reacting with OH radicals and ozone. The average ozone concentrations at the IAP site during sample collection were 13 ppb (WL), 6 ppb (WH), 39 ppb (SL) and 63 ppb (SH). In addition to ozone, the concentrations of other gas

pollutants as well as temperature and humidity data for the different samples can be found in the supplement in Tab. S2. For the BT sample, the lower O/C ratio likely reflects a largely primary particle composition.

Van Krevelen diagrams, in which H/C is plotted against O/C for each assigned formula (Kim et al., 2003), are widely used as a means to categorize aerosol samples since they provide a clear representation of the range of O/C ratios found in the organic aerosol sample (Nizkorodov et al., 2011). Van Krevelen plots for both the Beijing samples and the samples from Birmingham

are shown in Fig. 4. It can be seen that for the winter samples, a lot of formulae are located in the aromatic region (H/C<1, O/C<0.5) (Mazzoleni et al., 2012), and that there is a significant reduction of formulae in this region in the summer data. This aromatic region is also strongly populated in the BT sample. The SL period is nearly devoid of formulae in the aromatic region, while the aromatic region of SH is similar to the BB sample, which was recorded in early autumn. These observations about aromaticity are in agreement with the conclusions drawn from the total H/C ratio. The formulae that can be identified as

aromatic tend to be CHO and CHON. Conversely, CHOS and CHONS formulae are primarily located outside of this aromatic region, with much higher H/C ratios, indicating a lower level of aromaticity. Figure 4 shows that there is a visible increase in not just the percentage of sulphur-containing formulae (see Fig. 2), but also the absolute number of sulphurous formulae in WH over WL. A similar trend can be seen for the summer.

**Table 1: Average O/C and H/C ratios for each sample set.**

| Sample Set | Average O/C | Average H/C |
|---|---|---|
| Winter Low (WL) | 0.41 | 1.19 |
| Winter High (WH) | 0.48 | 1.25 |
| Summer Low (SL) | 0.54 | 1.53 |
| Summer High (SH) | 0.62 | 1.44 |
| Birmingham Background (BB) | 0.54 | 1.44 |
| Birmingham Tunnel (BT) | 0.42 | 1.16 |

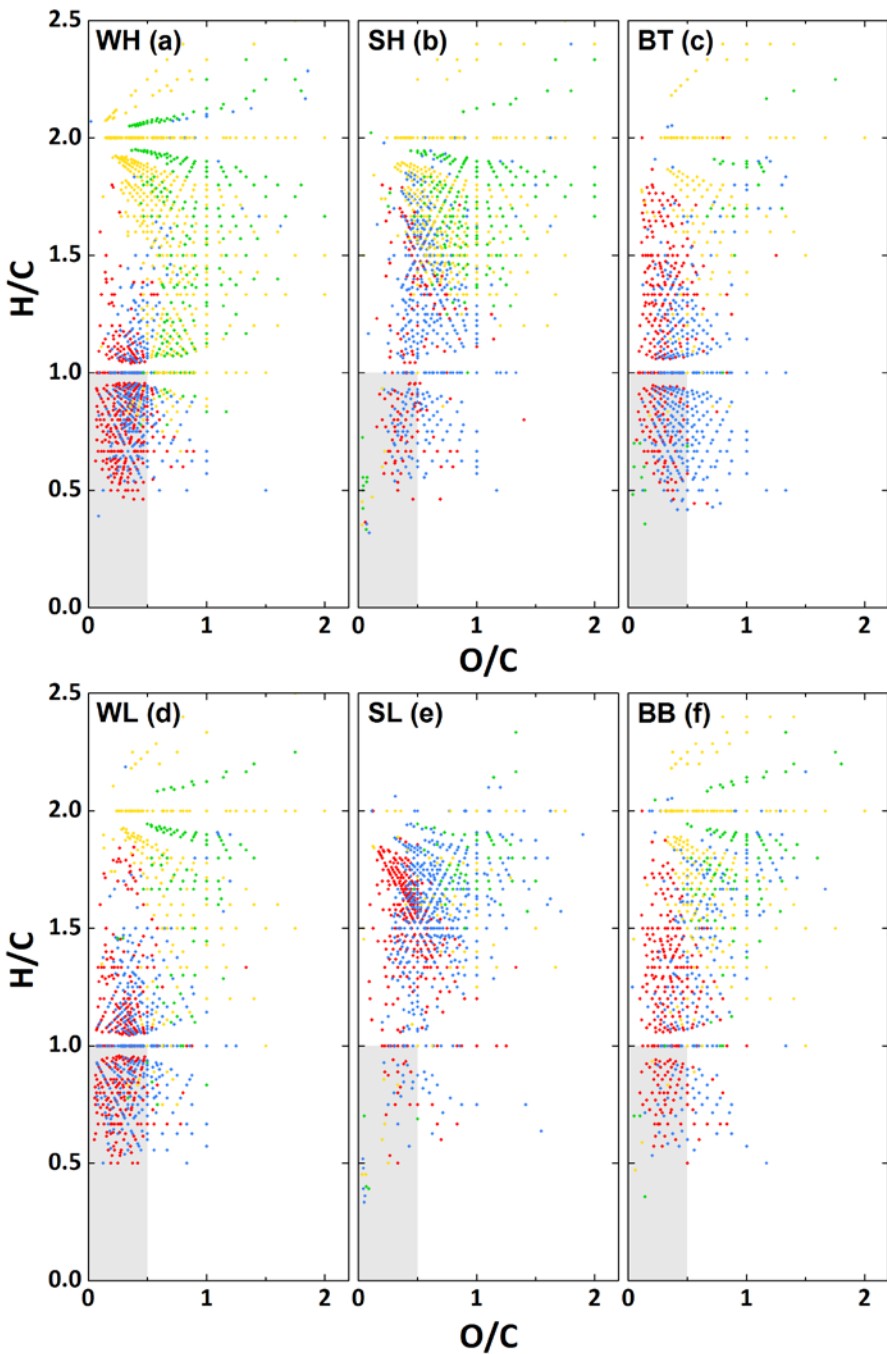

**Figure 4: Van Krevelen plot for the WH, WL, SH, SL, BT and BB sample sets. The colours denote the CHO (red), CHON (blue), CHOS (yellow) and CHONS (green) formulae detected for each sample set. The aromatic region is shaded in grey.**

## 3.2 Aromatic compounds

The low H/C ratios for several of the samples suggest the presence of a large number of aromatic compounds. Figure 5 shows a Van Krevelen plot for the different samples in which the different formulae are colour mapped according to their Double Bond Equivalent (DBE), which was calculated according to Eq. 1. The DBE gives the number of double bonds plus rings in a molecule. For DBE calculations, it is usally assumed that all atoms involved (except hydrogen) obey the octet rule. While this assumption tends to hold true in strictly reducing environments, under oxic conditions, such as in the atmosphere, it is likely that both sulfur and nitrogen compounds with higher valency are present. Such compounds for example include organosulfates or -nitrates and nitro compounds, which are frequently dectected in ambient particles. The DBE values calculated here therefore represent a lower limit. A high DBE indicates likely aromaticity of a compound. It can be seen that the region identified in Fig. 4 as containing most likely aromatic compounds, contains the formulae with the highest DBE. The smallest PAH, naphthalene, has a DBE of 7, thus all formulae in Fig. 5 shown in yellow, red or brown colours are probably polycyclic aromatics, likely oxidised PAHs. This assumption is corroborated by the findings of Elzein et al., (2019), who used gas chromatography–time-of-flight mass spectrometry to quantify the concentration of 10 oxygenated PAHs (OPAHs) and 9 nitrated PAHs (NPAHs) during the APHH winter campaign and found the total concentration of OPAHs to range from 1.8 to 95.5 ng·m$^{-3}$ and that of NPAHs from 0.13 to 6.43 ng·m$^{-3}$.

The DBE values of the molecules found in the aromatic region for WH and WL exceed that of the sample from BT. This suggests that larger polycyclic aromatics are found in Beijing air in the winter than in aerosol collected in a road tunnel in a European city, likely due to an increase in solid fuel burning for residential heating. A general increase in the concentration of PAHs and oxidised PAHs in China during winter, generally attributed to residential heating, has been observed in multiple studies (Bandowe et al., 2014). In addition to this, Zhang et al. (2017) found a sharp increase in the concentration of higher-ring-number (≥4) PAHs at the start of the heating season and Huang et al. (2014) observed a similar increase in emission of higher-ring-number PAHs for coal combustion compared to gasoline and diesel, lending weight to our hypothesis about the origin of the larger polycyclic aromatics.

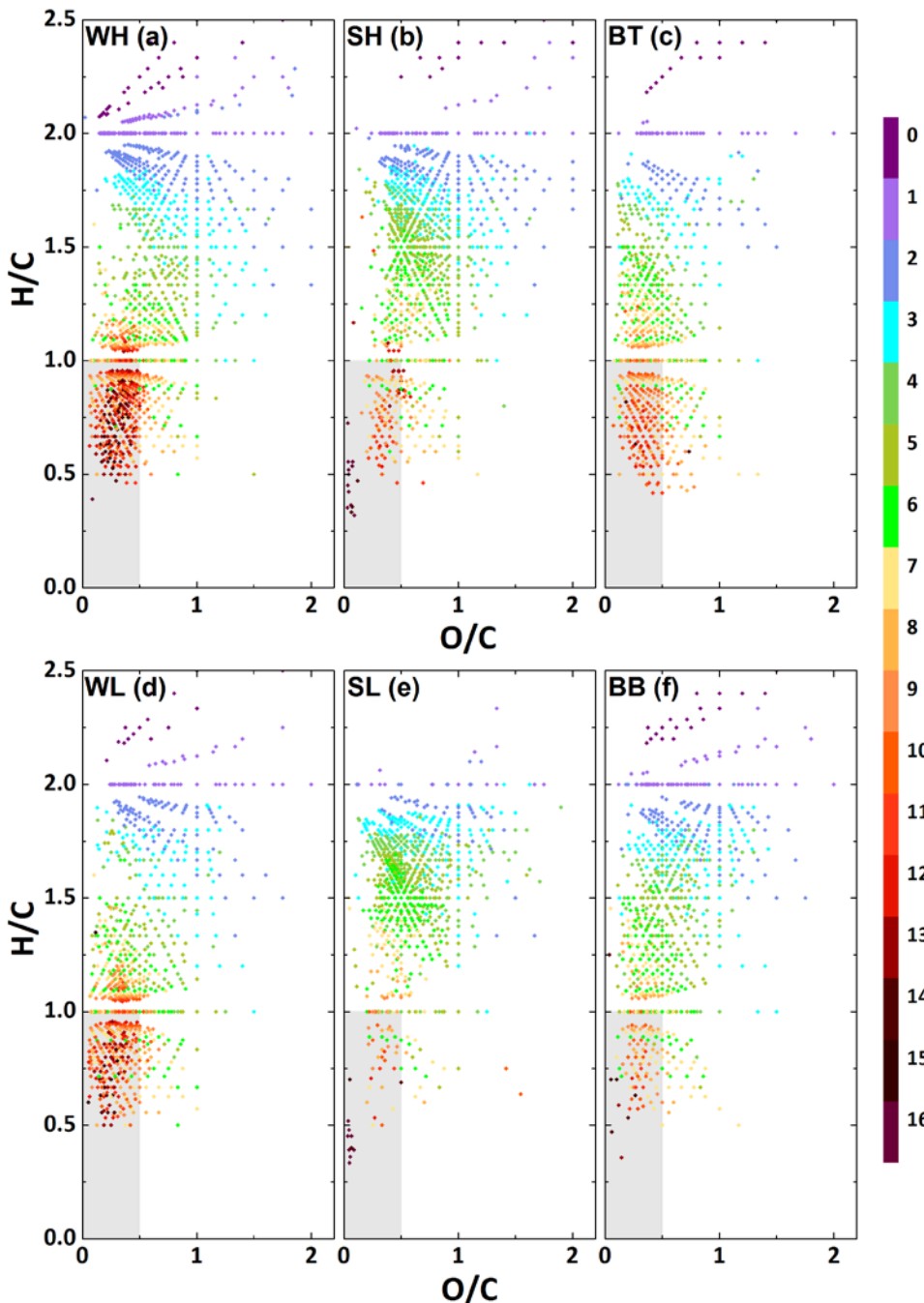

**Figure 5: Van Krevelen plot for the WH, WL, SH, SL, BT and BB sample sets. Colour indicates the Double Bond Equivalent (DBE), calculated under the assumption that sulfur is bivalent. The aromatic region is shaded in grey.**

While the DBE can be used to determine the number of C-C double bonds in pure hydrocarbons, heteroatoms present in a compound can form double bonds not contributing to aromaticity, ring formation or condensation (Koch and Dittmar, 2006). To overcome this problem associated with DBE another parameter has been developed, the aromaticity equivalent (Xc). For organic molecules containing only oxygen and/or nitrogen, the aromaticity equivalent can be calculated as follows (Yassine et al., 2014):

$$Xc = \frac{2C + N - H - 2mO}{DBE - mO} + 1$$

where C, N, H and O are the number of carbon, nitrogen, hydrogen and oxygen atoms respectively and m is the fraction of oxygen atoms involved in the π-bond structure of the compound which differs for different functional groups. For carboxylic acids, esters, and nitro functional groups m = 0.5. Since the measurements in this study were performed with electrospray ionisation in negative mode, carboxylic acids likely dominate the signal so m = 0.5 is presumed for the calculation of Xc. Organic peroxy compounds can account for a significant fraction of organic particulate matter. For these compounds m is <0.5, as less than half of the oxygen atoms are involved in a π-bond structure. If our samples contain a large number of peroxy compounds, we would be underestimating the degree of unsaturation in the sample. The same is true if most of the of nitrogen containing compounds are organonitrates rather than nitro compounds. The values should therefore be considered a conservative estimate. If DBE ≤ mO, then Xc = 0. Compounds with Xc ≤ 2.5000 are non-aromatic, for monocyclic aromatics 2.5000 ≤ Xc ≤ 2.7143 and for polycyclic aromatics Xc ≥ 2.7143. There are too many possible oxidation states of sulfur to correctly assign the aromaticity of the CHOS and CHONS compounds and so this characteristic was not investigated further for S-containing formulae.

The distribution of polycyclic, monocyclic aromatics and aliphatic formulae on a Van Krevelen plot for all samples is shown in Fig. 6. The vast majority of polycyclic aromatics can be found in the region outlined before as aromatic (H/C<1, O/C<0.5), however some have slightly higher H/C ratios, which may be due to extended alkyl chains as the H/C cut off does not account for this. The majority of monocyclic aromatics are found outside the aromatic region, which indicates that the majority of them contain long alkyl chains. Molecules classified as non-aromatic are found entirely outside of the previously defined aromatic region, as expected. The calculated Xc values confirm that the summer samples have many fewer aromatic formulae than the winter samples. The SL sample is particularly low in aromatic formulae (132 aromatic formulae vs. 801 in total), especially regarding polycyclic aromatics (26 formulae). In contrast, there are a reasonable number of polycyclic (85) and monocyclic (159) aromatics in SH. The two winter samples both show high contributions of aromatic formulae (76% in WH vs. 77% in WL of all detected formulae in these samples). However, the WH sample is strongly dominated by polycyclic aromatics with 403 polycyclic vs. 260 monocyclic aromatic formulae, while monocyclic and polycyclic aromatics are present in nearly equal numbers in the WL sample (320 monocyclic and 297 polycyclic aromatic formulae). The high fraction of aromatics present in the winter samples is consistent with the results from Wang et al. (2018), who also compared filter samples collected in Beijing wintertime air under high and low pollution conditions. As mentioned previously, a lot of the polycyclic aromatic compounds are likely oxidised PAHs. Oxidised PAHs such as nitro-PAH, oxy-PAH and hydroxy-PAH can be produced either directly

from incomplete combustion or pyrolysis of fossil fuel and biomass or through oxidation of PAHs in the atmosphere (Albinet et al., 2007; Andreou and Rapsomanikis, 2009; Atkinson and Arey, 1994; Bandowe and Meusel, 2017; Walgraeve et al., 2010). Bandowe et al. (2014) found higher concentrations of oxidised PAHs in winter in Xi'an, a Chinese megacity. They partially attribute this increase to heating activities. A similar increase in the concentration of oxidised PAHs during heating season was observed in Beijing by Lin et al., (2015), who also state that this increase might be linked to winter heating activities. Their source apportionment showed that oxy- and hydroxy-PAHs in particular are dominated by biomass burning emissions during heating season. They state that during the non-heating period, secondary sources become more relevant due to increased photochemical activity, dominating as a source for oxy-and nitro PAHs. This is consistent with our observations of increased O/C during the summer (Table 1). Studies by Liu et al. (2019) and Lyu et al. (2019) confirm that coal combustion and biomass burning where significant sources of particulate matter during the APHH winter campaign, while a quantitative comparison with the summer is still pending.

To further investigate the different aromatic compounds, the approximate average carbon oxidation state ($\overline{OS_C}$) for each formula was plotted against the number of carbon atoms ($n_C$), with all formulae classified as polycyclic aromatics, monocyclic aromatics or non-aromatic Results for CHO compounds are shown in Fig.7, while results for the CHON compounds can be found in the supplement (Fig. S3). $\overline{OS_C}$ was calculated for each molecule as follows (Kroll et al., 2011):

$$\overline{OS_C} \approx 2\,O/C - H/C$$

where O, C and H are the number of oxygen, carbon and hydrogen atoms respectively . $\overline{OS_C}$ is used as an alternative metric to O/C for assessing the degree of oxidation since the O/C ratio of organic compounds can also change when a compound undergoes non-oxidative reactions. As different formulae can have the same combination of $\overline{OS_C}$ and carbon number, Fig. 7 and S3 have a lot of overlapping points. In Fig. S4, an offset was applied to the overlapping points to make more data points visible. Figure 7 shows that the summer samples contain large numbers of aliphatic CHO formulae with a huge range of different carbon numbers, whereas the non-aromatic formulae in the winter samples only rarely have more than 15 carbons. This is possibly due to increased biogenic emissions of long-chain fatty acids, alkenes and similar compounds during summer. The majority of the polycyclic aromatics and monocyclic aromatics in the summer are below Carbon Number of 20 (C20). Similarly, the BT sample, where road traffic is the main source, contains hardly any polycyclic aromatics above C18 and only few monocyclic aromatics above C20. This is in strong contrast with the winter samples from Beijing, which show a strong presence of polycyclic aromatics with up to 25 carbon atoms. The winter samples contain large numbers of aromatics, especially polycyclic aromatics above C15. The likely source for these is heating, as they are present in large numbers in both WH and WL yet not in the summer. This link between higher-ring number polyaromatics and residential heating is supported by studies showing a sharp increase in the concentration of higher-ring-number PAHs at the start of the heating season in Beijing (Zhang et al., 2017a) and increased emissions of higher-ring-number PAHs for coal combustion compared to gasoline and diesel (Huang et al., 2014). Similar trends can be observed for the CHON compounds in Fig. S3.

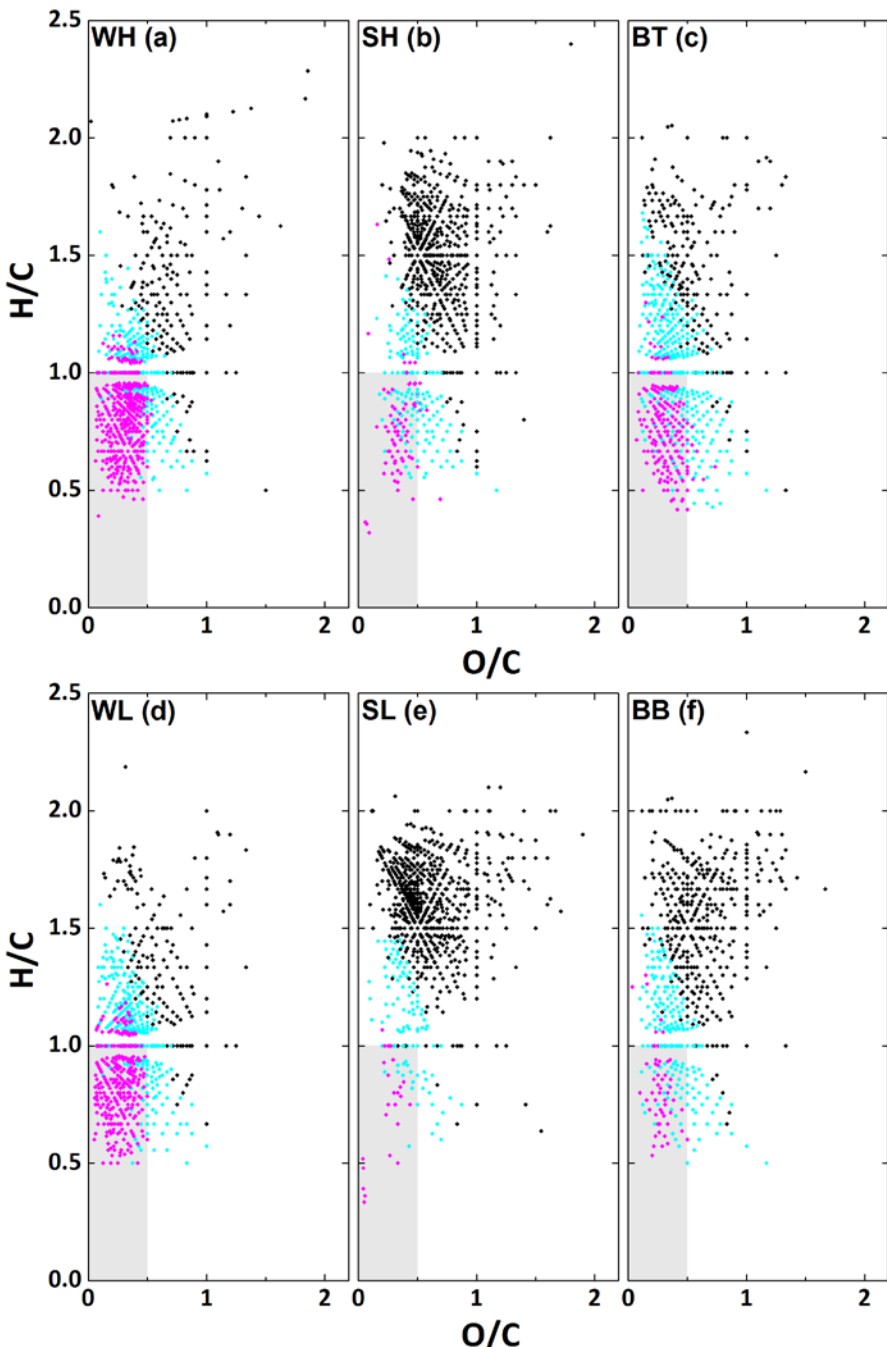

**Figure 6: Van Krevelen plot for the WH, WL, SH, SL, BT and BB sample sets. Polycyclic aromatic (2.7143 ≤ Xc), monocyclic-aromatic (2.5000 ≤ Xc ≤ 2.7143) and non-aromatic (Xc ≤ 2.5000) formulae for all CHO and CHON ions present in the samples are indicated in magenta, cyan and black, respectively. The aromatic region is shaded in grey.**

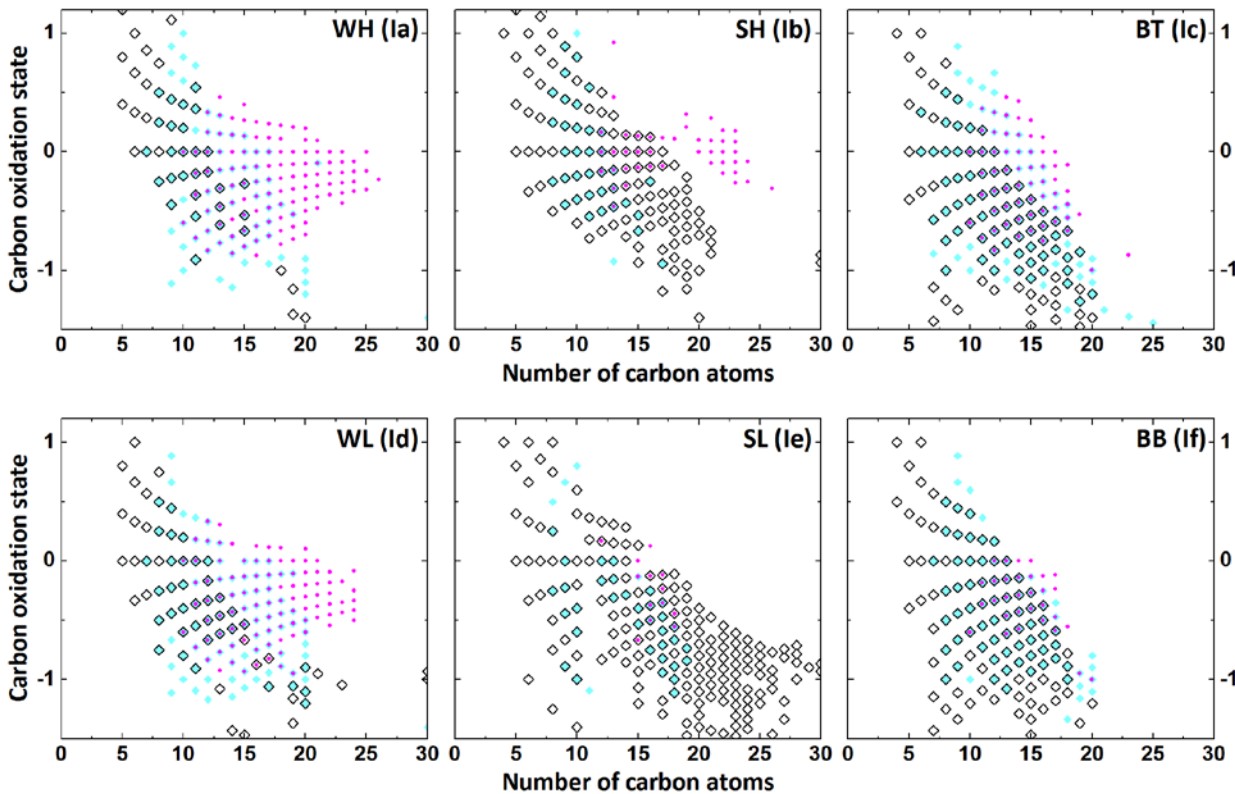

**Figure 7: Plot of carbon oxidation state against carbon number for all CHO formulae in the WH, WL SH, SL, BT and BB samples. Polycyclic aromatic (2.7143 ≤ Xc), monocyclic aromatic (2.5000 ≤ Xc ≤ 2.7143) and non-aromatic (Xc ≤ 2.5000) formulae are depicted as magenta circles, cyan diamonds and open black diamonds respectively.**

## 3.3 Sulfur- and Nitrogen-Containing Compounds

The mass spectra of all four investigated conditions show that the majority of assigned formulae contain sulfur and/or nitrogen atoms. Organic compounds containing sulfur and nitrogen such as organosulfates, organosulfonates, nitrooxyorganosulfates, amines, organonitrates and nitro compounds are prevalent in the atmosphere (Bandowe et al., 2014; Canales et al., 2018; Huang et al., 2012b; Iinuma et al., 2007; Kiendler-Scharr et al., 2016; Kristensen and Glasius, 2011; Lee et al., 2016; Riva et al., 2015; Valle-Hernández et al., 2010). As mentioned previously, all ionisation methods for mass spectrometry differ in their ionisation efficiency towards different compounds. Nonpolar compounds are generally not ionised well in ESI and negative mode ESI is not sensitive towards reduced nitrogen compounds such as amines and imines, while organosulfates, organosulfonates, organonitrates and nitro compounds should ionise well.

Oganosulfates are estimated to contribute as much as 30% to organic fine particle mass (Surratt et al., 2008). Secondary organosulfate formation likely proceeds through particle-phase chemistry via a variety of proposed mechanisms such as esterification of hydroxyl and keto groups (Liggio and Li, 2006), acid-catalysed ring-opening of epoxides (Iinuma et al., 2009;

Minerath and Elrod, 2009), radical-initiated processes (Galloway et al., 2009; Nozière et al., 2010) and the direct reaction of $SO_2$ with unsaturated compounds (Passananti et al., 2016). Inorganic $SO_4^{2-}$ in particles is formed through oxidation of sulfur dioxide, which is primarily emitted from anthropogenic sources, especially over land (Smith et al., 2001). In Beijing, $SO_2$ concentrations tend to be particularly high in winter (Zhang et al., 2011; Zhou et al., 2015), with a large contribution from

heating (Huang et al., 2012a). However, the resulting concentration of $SO_4^{2-}$ also strongly depends on the rate of $SO_2$ oxidation. In Beijing, average $SO_4^{2-}$ concentrations in fine particles are usually higher in summer than in winter (Chen et al., 2017; Hu et al., 2016; Huang et al., 2016; Liu et al., 2017), which is attributed to increased photooxidation during summer. In our study, the average sulfate concentration was however slightly lower in summer (6.9 $\mu g/m^3$) than in winter (8.5 $\mu g/m^3$). This unusually low concentration compared to the cites studies might be due to the fact that our campaign was conducted earlier in summer

than the others. Studies have also shown that the formation rate of $SO_4^{2-}$ is particularly high during haze episodes, with a strong contribution from heterogeneous oxidation within haze droplets (Ma et al., 2018; Wang et al., 2006), leading to high maximum concentrations of sulfate during haze events, of which several strong ones occurred in winter during our campaign. While organosulfate formation has been primarily understood to be a secondary process, a recent direct infusion UHRMS study of coal combustion by Song et al. (2019) found 5-25 % of formulae in the methanol-extracted fraction to contain sulfur, indicating

the potential importance of direct organosulfate emissions. A study by Wang et al. (2019) showed that organosulfates in wintertime $PM_{2.5}$ in Beijing originated from multiple types of biogenic and anthropogenic precursors.

The majority of S-containing formulae in our samples contain only one sulfur atom. Out of those, most (≥99% for WH and WL, >92% for SH and SL) also contain at least four oxygen atoms, marking them as potential organosulfates. This is in line with previous studies (Jiang et al., 2016; Tao et al., 2014; Wang et al., 2019). For the CHONS formulae, 77% in WH, 94% in

WL, 86% in SH and 91% in SL have at least 7 oxygen atoms, indicating potential nitrooxy organosulfates.

Unlike organosulfates, formation of organonitrates is thought to occur mostly in gas phase through e.g. the reaction of peroxy radicals with NO or oxidation of volatile organic compounds through the $NO_3$ radical (Ng et al., 2017; Zhang et al., 2004). Inorganic $NO_3^-$ in particles is predominantly present in the form of $NH_4NO_3$, which is formed through reaction of $NH_3$ with $HNO_3$. The dominant pathways for $HNO_3$ formation are the reaction of $NO_2$ with OH radical during daytime and $N_2O_5$

hydrolysis during the night (Bauer et al., 2007; Khoder, 2002). In Beijing, ammonium-poor particles were found to still have very high $NO_3^-$ content. It has been suggested that hydrolysis of $N_2O_5$ in particles is responsible for this phenomenon (Pathak et al., 2009). Just like for $SO_4^{2-}$, $NO_3^-$ concentrations in Beijing are enhanced during haze episodes (Huang et al., 2016; Wang et al., 2006). Apart from organonitrates, nitroaromatics are another important class of N-containing organics in the atmosphere which can be detected negative mode ESI. They can either be emitted directly via combustion of biomass and fossil fuels

(Heeb et al., 2008; Karavalakis et al., 2010; Wang et al., 2017b) or formed in the atmosphere through reaction of aromatics (Keyte et al., 2013). Apart from being both a primary and secondary source of nitroaromatics, biomass burning can also lead

to formation of other nitrogen-containing organics, such as alkaloids (Laskin et al., 2009). However, these compounds are unlikely to be detected in negative mode ESI.

The majority of detected N-containing formulae (97% WH, 89% WL, 98% SH and 97% SL) also contained at least three oxygen atoms, marking them as potential organonitrates. Nitro compounds on the other hand have a minimum of two oxygens

per molecule, which is the case for >99% of N-containing formulae in the WH, SH and SL samples and 96% in the WL sample.

To obtain further information about the potential sources of the sulphur- and nitrogen-containing compounds in our study, the total number of molecular formulae containing sulphur and/or nitrogen were plotted against the concentration of inorganic sulphate ($SO_4^{2-}$) and nitrate ($NO_3^-$) (Fig. 8) for each measured sample respectively. The direct infusion MS analyses done here

does not allow to determine compound concentrations accurately, thus we use the number of CHOS compounds as an indicator for the importance of the formation processes for S-containing organic compounds. An overview of the correlation paramenters can be found in the supplement (Tab. S4). For the winter samples, there was a statistically significant positive correlation between the number of CHOS formulae and the concentration of inorganic $SO_4^{2-}$ on that day (Fig. 8a), corroborating results of other studies that S-containing organics are formed in the particle phase. The sulphate data did not correlate as well for

summer (Fig. 8b), which might be explained by the lower maximum concentrations of $SO_4^{2-}$ and the on average slightly lower $SO_4^{2-}$ concentrations during our summer campaign, where particle-phase formation reactions of S-containing organics might become less important. Very similar trends are observed for CHONS, suggesting common formation routes for these compounds as for CHOS. The ions that are found only in the WH and not the WL sample are likely to be representative of winter haze events. Over half of these components are sulphur containing (Fig. S1) – which suggests high formation rates for

organic sulphurous compounds during winter haze events. Wang et al. (2019) found that organosulfates with a high carbon number were significantly more abundant in polluted Beijing winter samples than in samples taken under low pollution conditions, which indicates that in polluted air, more OSs are generated from long-chain alkanes.

It can be seen in panel c and d of Fig. 8 that the number of CHON formulae does not correlate significantly with the $NO_3^-$

concentration while CHONS does, though this correlation is only statistically significant for the winter samples. This suggests that in contrast to the S-containing organics, most N-containing organics detected in our samples are not formed in the particle phase. As stated earlier, the detection mode we used is biased towards oxidised N-containing organics such as organic nitrates, which are known to form predominantly in the gas phase and and nitro compounds, which can be of either primary or secondary origin, with a strong contribution of gas-phase oxidation in the second case. A correlation with particle-phase nitrate is

therefore not expected.

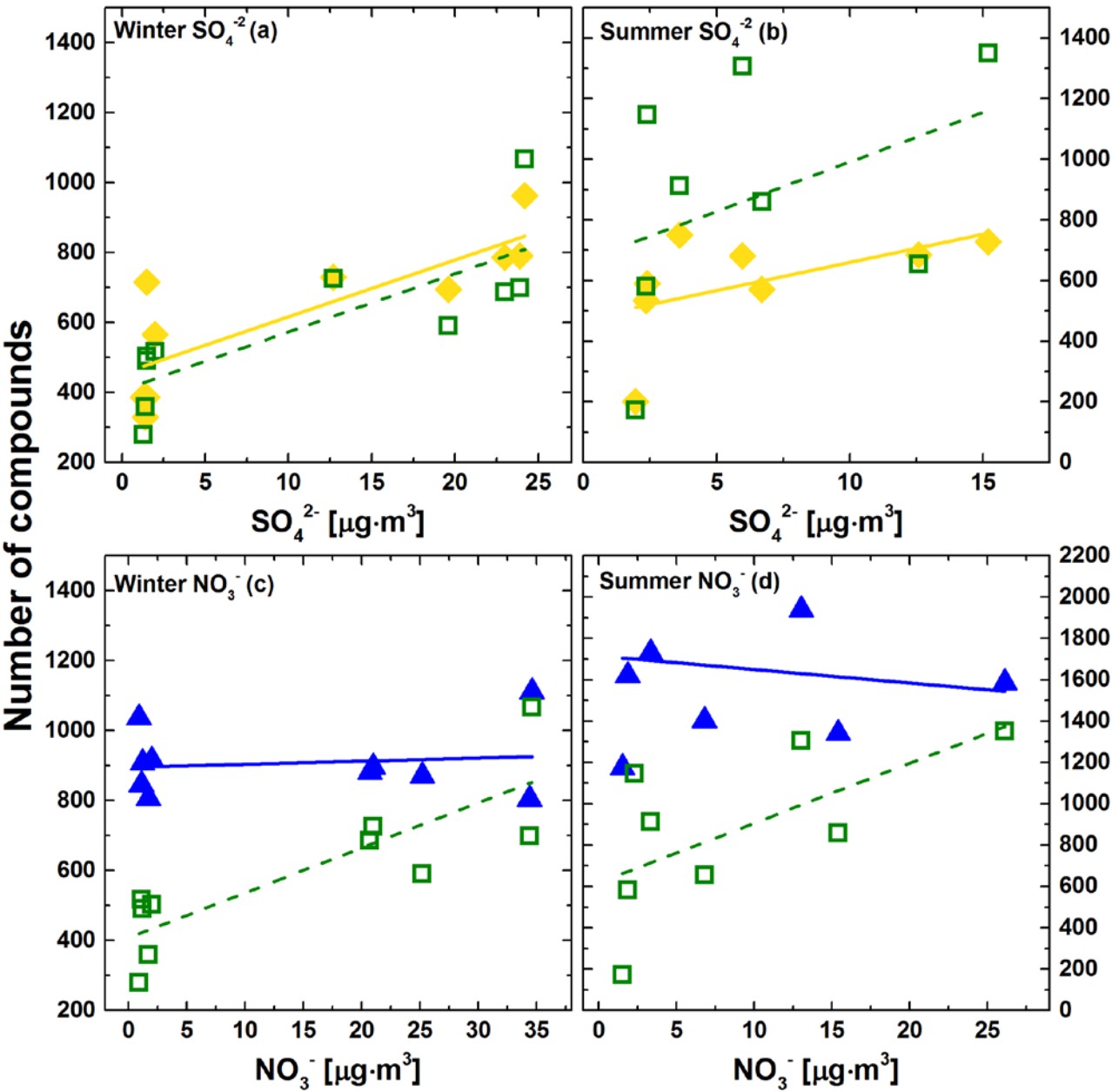

**Figure 8: Number of molecular formula with the elemental composition CHOS (yellow diamonds), CHONS (green square) and CHON (blue triangles) plotted against the concentration of SO$_4^{2-}$ (a&b) or NO$_3^-$ (c&d) in the particle phase, analysed via ion chromatography. An overview of the correlation paramenters can be found in the supplement (Tab. S4).**

## Conclusions

Comparison of $PM_{2.5}$ filter samples from Beijing from the winter and summer APHH-Beijing campaign were analysed with direct injection ultra-high resolution mass spectrometry. A strong variation in the composition of organic particles was observed between high and low pollutiuon conditions conditions as well as between the two seasons. In summer significantly more CHON formulae were detected compared to winter, likely due to the increased NOx-driven photochemistry in summer. S-containing formulae were more dominant during high-pollution events in both seasons and highest during winter haze events. The contribution of aromatic compounds was strongly increased in winter, likely due to heating as an additional source. The relative contribution of S-containing formulae increased under high pollution conditions and the number of S-containing formulae showed correlation with inorganic $SO_4^{2-}$ on the filters, consistent with a particle-phase formation process, while no such correlation was found for N-containing formulae and inorganic $NO_3^-$.

## Data availability

The direct infusion ultra-high resolution mass spectrometry data is available through the CEDA Archive (https://catalogue.ceda.ac.uk/uuid/680ebdcb83c244fdb9d069e2f8952812).

## Author contributions

MK and SSS designed the research. DJP and SSS performed the mass spectrometry measurements and data analysis. TVV measured the ion chromatography data. MP ran the NAME model. SSS wrote the manuscript with contributions from all co-authors.

## Competing Interests

The authors declare that they have no conflict of interest.

## Special issue statement.

This article is part of the special issue "In-depth study of air pollution sources and processes within Beijing and its surrounding region (APHH-Beijing) (ACP/AMT inter-journal SI)". It is not associated with a conference.

## Acknowledgement

We acknowledge the support from Pingqing Fu, Zifa Wang, Jie Li and Yele Sun from IAP for hosting the APHH-Beijing campaign at IAP. We thank Di Liu and Bill Bloss from the University of Birmingham, Siyao Yue, Liangfang Wei, Hong Ren,

Qiaorong Xie, Wanyu Zhao, Linjie Li, Ping Li, Shengjie Hou, Qingqing Wang from IAP, Rachel Dunmore, Ally Lewis and James Lee from the University of York, Kebin He and Xiaoting Cheng from Tsinghua University, and James Allan and Hugh Coe from the University of Manchester for providing logistic and scientific support for the field campaigns.

This research was funded by the UK Natural Environment Research Council (NERC) as part of the APHH-Beijing study (NE/N007190/1 and NE/N007158/1). SSS acknowledges support from the Swiss National Science Foundation (Project No. 162258) and a 2017 LIFE PostDoc fellowship by the AXA Research Fund.

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
