# Peer review of "Differences in the Composition of Organic Aerosols between Winter and Summer in Beijing: a Study by Direct Infusion Ultrahigh Resolution Mass Spectrometry"

_Atmospheric Chemistry and Physics, 2019_

## Referee Comment (RC1) · Anonymous Referee #2 · 31 Jan 2020

General comments:

This paper characterized the elemental composition of polar organic compounds in PM2.5 from Beijing during wintertime and summertime. 918-1586 organic compounds were assigned molecular formulae by direct infusion negative nano-electrospray ionization ultrahigh resolution mass spectrometer. Then the differences of the chemical composition under different pollution conditions, and their potential primary and secondary sources were discussed. The overall strength of this study is acquisition of a detailed dataset of polar organic compounds that spanned summertime and wintertime, low and high pollution conditions. The topic of the paper is well suited for ACP,

and the data itself are interesting. On the whole, the language requires improvement throughout the manuscript. Many sentences are not clearly written, leaving the reader puzzling about their meaning. In addition, the overall weakness is the data interpretation. Much more effort needs to be put into presentation of the results. I have some points where more information is needed or where I disagree.

I use the abstract to illustrate my main concerns of this paper. The authors state in the abstract: . . ...There were strong differences in aerosol particle composition between the seasons. . ...which were strongly enhanced in winter, likely due to increased fossil fuel and biomass burning for heating. . . ...the contribution of sulfur-containing organic compounds strongly enhanced under high pollution conditions. . . ....

The highlighted results above are not really exciting. In fact, many researches have reported these differences already. For the advantage of using the ultrahigh resolution mass spectrometer, I suggest the authors to focus on the new findings of this study.

Specific comments:

1. Introduction: The characterization and source identification of organic compounds in PM in Beijing have been extensively studied. I would suggest authors to improve the introduction by summarizing these previous studies and providing some results in line with the major conclusion of this study.

2. Page 2, Line 10-12: "As a central of the project,. . . .." It makes no senses for your paper since the information on sampling has been provided in the Section Materials and Method.

3. Page 3: Maybe the meteorological conditions (e.g. temperature, RH, wind speed, wind direction, mixing layer height, etc.) play an important role on the organic components. I would suggest the authors to give a general characteristics of gas and PM pollutants and meteorological conditions during low and high pollution conditions (WH, WL, SH, SL), respectively, since the authors focus on the comparison between the

characteristics of organic groups under different pollution conditions, in the Supplementary Information, for instance?

4. Page 4, Line 16: What does "0.3âĽęH/CâĽğ2.5" mean?

5. Page 4, Line 21: Here x, y and z are the number of carbon, hydrogen, nitrogen atoms. However, in Page 12, Line 7, C, N, H and O are the number of carbon, nitrogen, hydrogen and oxygen. I suggest the authors keep consistent in the nomination to avoid the confusion.

6. Page 5, Lines 17-20: It makes no senses since the air masses have been discussed in your paper.

7. Page 5, Lines 22-25: Here these sentences make no sense for the paper.

8. Page 5: Here DBE was calculated based on the number of C, H, N atoms. I am not sure whether DBE should be calculated with the relative abundance weight since the relative abundance of each molecule in the mass spectrum is different. The same consideration also apply for the other parameters such as H/C, O/C, Xc and so on.

9. Page 6, Line 5: A reference would be helpful.

10. Page 6: I suggest the authors add the number of four different groups identified in different pollution conditions in Figure 1.

11. Pages 6-8: The chemical composition and some ratios (i.e., O/C, H/C) of organic compounds have been characterized by ultrahigh resolution mass spectrometer in some cities such as Beijing and Shanghai, and some typical emission sources (i.e., coal combustion, biomass burning). I suggest the authors compare their result with those reported in literature.

12. Page 7, Line 14: You are right, here a reference would be helpful.

13. Page 7, Line 15: What does "high and low H/C particulate matter" mean?

[Figure]

14. Page 8, Lines 1-2: The SH showed high H/C ratio. The authors suggest that is due to a large proportion of biogenic organic aerosol from plant sources. It might be good here to give more evidences. Is there any assigned formula which could be used as markers for biomass burning or biogenic organic aerosol? For example, nitrophenols, nitrocatechols? They show higher number fraction and/or relative abundance in SH samples?

15. Page 8, Line 9: It might be better to give the ozone concentrations for the WH, WL, SH, SL samples.

16. Page 9: The VK plots show the aromaticity of CHO and CHON is quite different in winter and summer. Please speculate more in depth on the difference.

17. Page 10: Here the authors give some data on polyaromatic compounds. How about the single-ring aromatics? I think they contribute more to the aromatic compounds.

18. Page 12: The authors spend too many words to discuss the aromaticity equivalent (Xc). I suggest the authors focus on the new finding which cannot be deduced from the H/C and O/C ratios.

19. Page 13, Line 21: A reference regarding the heating source would be helpful.

20. Page 16, Line1: The authors state that sulfate concentrations are usually higher in summer than in winter in Beijing. Are you sure? Please give the concentrations of sulfate and nitrate in WH, WL, SH, and SL samples in a Table, for instance, in the Supplementary Information. And in Line 25, you state the lower sulfate concentrations in summer. It is significantly contradictory.

21. Page16: It is a good idea to discuss the relationship between sulfate and nitrate with the number of CHOS and CHON compounds, but I recommend the authors to provide in-depth insights into this discussion. In addition to the secondary formation, Song et al. (EST 2019, 53, 13607-13617; 52, 2575-2585) reported that S-containing compounds account for 36% of the total number of compounds identified, making up

the largest component in coal smoke, and N-containing compounds show high abundance in biomass burning. The primary sources of S- and N-containing compounds should also be considered.

22. Page 16: The authors state that the particles in winter are sampled before they can undergo atmospheric ageing processes, for example reacting with OH radicals and ozone in Page 8, Lines 6-7. It seems inconsistent with the good positive correlation between sulfate and CHOS compounds.

Technical corrections:

1. Page 5, Lines 27-28: What does "Fehler! Verweisquelle konnte nicht gefunden warden" mean? Mistype?

2. Page 5, Line 30: "off" should be "of".

---

## Referee Comment (RC2) · Anonymous Referee #3 · 23 Apr 2020

This study reports the analysis of organic aerosol composition in central Beijing during winter and summer months using orbitrap mass spectrometry coupled with a nano-electrospray ionisation source. A main finding is that the number of S-containing organic species increased with inorganic sulfate concentration. This work provides new information on organic aerosol chemistry in northern China with a scope that fits well within ACP. But there are some issues on data analysis and interpretation, which may require major revisions to resolve.

It is important that the authors discuss the limitations with negative ESI MS analysis, such as its low ionization efficiency towards, or inability to detect, certain types of

compounds. The calculation of aromaticity equivalent Xc is based on assumptions of elements' valences which may not always hold for atmospheric organics. The authors excluded S-containing species in their calculations of Xc, but nevertheless calculated the Xc for N-containing compounds, in which the valence of N can be either 3 or 5. In addition, the O valence in peroxides is 1 rather than 2 and it is known that organic peroxides can account for a significant fraction of the molecules. The authors should discuss how these issues affect their results and conclusions. Additionally, some discussions on the technical aspects a bit vague and need clarification. See below for specific comments.

This study only compares winter and summer compositional differences, thus is an overstatement to have a title of "Seasonal Differences . . ."

Page 2, how was sampling from Birminghan UK decided to be representative of a typical European urban background site?

Page 3, What's the sampling duration for the filters?

Page 3, Line 19, is the concentration corresponding to PM mass or OA mass? How was it known?

Page 4: • Line 1-2, what's the mass accuracy of the instrument? • Line 7 – 9, this sentence is vague. More information is needed to clarify how this was done. • Line 11, be specific about the threshold to remove signals and define how noises are determined. • Line 12, "blank subtraction" usually means that all ions were subjected to blank subtraction, but this sentence suggests only the ions less than 10 times of the blank level are removed. This is confusing. • Line 20 -21, the formula for DBE calculation has limitations due to the assumptions about element valences. This issue should be clearly stated and the implications on the reported results should be discussed. • Line 23 – 24, the sentence "If . . ." is confusing. Please be specific.

Page 5: • The Panagi et al. paper is not yet published and unavailable. It is not

appropriate to cite it as a source of information used in this paper. Either provide the paper as supplementary or reiterate relevant key points. • The meaning of "the residence time of the air masses (or the integrated concentration of theoretical air mass particles)" is not straightforward, needs clarification. • Line 22-24 seems unnecessary, consider to remove. • There are strange characters shown at Line 27 – 28. • Change "off" to "of" on Line 30

Page 7. Line 14. Ref?

Page 8: • Line 1-2, waxy biogenic organic aerosol components likely have high H/C, but not all biogenic compounds have high H/C. It is more useful to define the cutoff value of "high H/C" • line 14, what's the basis for claiming that compounds with H/C < 1 and O/C < 0.5 are aromatic? Citing a previous study here without proper context is not sufficient.

Page 10, specify the "low" and "high" values use in describing elemental ratios and discussing chemical meanings.

Page 12, 1st paragraph, N also has two valences, so what's the validity of calculating Xc for N-containing compounds using the given formula?

Page 16, line 33, what's the reasoning behind this sentence – "This suggests ..."? Why does the correlation suggest how the compounds are formed? The authors appear to imply that the N-containing ions detected in this study are representative of " N-containing organics" in aerosol, but this is misleading as negative mode ESI-MS generally biases against reduced nitrogen compounds. Such issues should be articulated throughout the manuscript.

---

## Author Comment (AC1) · 20 Jul 2020

The authors thank the reviewer for taking the time to review this manuscript and for the constructive criticism.
This document includes authors' responses to anonymous referee #2 (RC1). Reviewer's comments are in black text while the authors' responses are in blue, with amended text quoted from the manuscript in quotation marks. Page numbers and lines refer to the revised version unless explicitly stated otherwise.

**Anonymous Referee #2 (RC1)**

*General comments:*

This paper characterized the elemental composition of polar organic compounds in PM2.5 from Beijing during wintertime and summertime. 918-1586 organic compounds were assigned molecular formulae by direct infusion negative nano-electrospray ionization ultrahigh resolution mass spectrometer. Then the differences of the chemical composition under different pollution conditions, and their potential primary and secondary sources were discussed. The overall strength of this study is acquisition of a detailed dataset of polar organic compounds that spanned summertime and wintertime, low and high pollution conditions. The topic of the paper is well suited for ACP, and the data itself are interesting. On the whole, the language requires improvement throughout the manuscript. Many sentences are not clearly written, leaving the reader puzzling about their meaning. In addition, the overall weakness is the data interpretation. Much more effort needs to be put into presentation of the results. I have some points where more information is needed or where I disagree. I use the abstract to illustrate my main concerns of this paper. The authors state in the abstract: . . .. . .There were strong differences in aerosol particle composition between the seasons. . ...which were strongly enhanced in winter, likely due to increased fossil fuel and biomass burning for heating. . .. . .the contribution of sulfur-containing organic compounds strongly enhanced under high pollution conditions. . .. . ..The highlighted results above are not really exciting. In fact, many researches have reported these differences already. For the advantage of using the ultrahigh resolution mass spectrometer, I suggest the authors to focus on the new findings of this study.
We recognise that not all results presented here are absolutely new but we strongly believe that they are worth reporting as they clearly illustrate the organic compositional differences in Beijing between seasons and pollution levels.

*Specific comments:*

1. Introduction: The characterization and source identification of organic compounds in PM in Beijing have been extensively studied. I would suggest authors to improve the introduction by summarizing these previous studies and providing some results in line with the major conclusion of this study.
This part of the introduction section has been expanded and we added additional references (page 2, line 27-page 3, line 2).

2. Page 2, Line 10-12: "As a central of the project,. . .. . ." It makes no senses for your paper since the information on sampling has been provided in the Section Materials and Method.
This paper is part of the APHH campaign, which is a central piece of information for this paper. Thus, we believe it is appropriate to mention this at the very start of the paper (very shortly; 1.5 lines), as well as more detailed in the methods section.

3. Page 3: Maybe the meteorological conditions (e.g. temperature, RH, wind speed, wind direction, mixing layer height, etc.) play an important role on the organic components. I would suggest the authors to give a general characteristics of gas and PM pollutants and meteorological conditions during low and high pollution conditions (WH, WL, SH, SL), respectively, since the authors focus on the

comparison between the characteristics of organic groups under different pollution conditions, in the Supplementary Information, for instance?

The available information regarding gas-phase pollutants and meteorological conditions (RH, T) has been added to the supplementary information in table S2 and the following sentence has been added on page 10, line 9:

"In addition to ozone, the concentrations of other gas pollutants as well as temperature and humidity data for the different samples can be found in the supplement in Tab. S2."

The information one could get from wind direction regarding the particle sources is very similar as discussed in the back trajectory section. Boundary layer heights during this campaign are unfortunately currently not available to us.

4. Page 4, Line 16: What does "0.3âL' ̧eH/CâL' ˘g2.5" mean?

These symbols are displayed properly in the PDF we downloaded. We will make sure that they are displayed correctly in the final version.

5. Page 4, Line 21: Here x, y and z are the number of carbon, hydrogen, nitrogen atoms. However, in Page 12, Line 7, C, N, H and O are the number of carbon, nitrogen, hydrogen and oxygen. I suggest the authors keep consistent in the nomination to avoid the confusion.

The formula for calculation of the double bond equivalent was using different nomination since it is also valid for other elements with the same valence. That section has now been reworded to clarify this (p. 4, Line 31):

"with $I_y II_n III_z IV_x$, where I=monovalent elements; II=bivalent elements; III=trivalent elements and IV=tetravalent elements. Sulfur was assumed to be bivalent and nitrogen trivalent for this calculation."

6. Page 5, Lines 17-20: It makes no senses since the air masses have been discussed in your paper.

Unlike the data shown in this manuscript, the referenced paper also covers time periods outside the duration of the APHH campaign. The sentence has been changed as following to state this more explicitly (p. 5, Line 27):

"More in-depth information about the origin of different air masses in Beijing, including time periods not covered by the APHH campaign, can be found in Panagi et al. (2020)."

7. Page 5, Lines 22-25: Here these sentences make no sense for the paper.

These sentences were removed as suggested by the reviewer.

8. Page 5: Here DBE was calculated based on the number of C, H, N atoms. I am not sure whether DBE should be calculated with the relative abundance weight since the relative abundance of each molecule in the mass spectrum is different. The same consideration also applies for the other parameters such as H/C, O/C, Xc and so on.

While a weighting calculation based on signal intensity would be possible for averaged metrics such as O/C and H/C ratio, we have come to the conclusion that it would not be more appropriate than the current method since signal abundance cannot be directly linked to compound concentrations due to matrix effects and variation in ionisation efficiency.

9. Page 6, Line 5: A reference would be helpful.

The text has been changed to (p. 6, line 11):

A trend that can be seen in the detected formulae is the presence of significantly more CHON formulae in summer compared to winter. The opposite trend was observed in Shanghai by Wang et al. (2017a), who found higher numbers and relative contribution of CHON in winter. Further information would be needed to explain this discrepancy.

10. Page 6: I suggest the authors add the number of four different groups identified in different pollution conditions in Figure 1.

We have now added the absolute numbers of detected formulae for the different groups for each sample in figure 2 (formerly figure 1). No other changes were made to the content of the figure, although we have changed the order of the samples to bring it in line with the other figures.

11. Pages 6-8: The chemical composition and some ratios (i.e., O/C, H/C) of organic compounds have been characterized by ultrahigh resolution mass spectrometer in some cities such as Beijing and Shanghai, and some typical emission sources (i.e., coal combustion, biomass burning). I suggest the authors compare their result with those reported in literature.

While there are several studies looking at chemical composition of organic compounds in PM in Chinese megacities, results are often difficult to compare, especially quantitatively, due to differences in methodology. The extraction solvent can for example strongly influence the chemical composition (see e.g. the Song et al. reference suggested by the reviewer), with many studies using other solvents than methanol, which we used in our study. In addition, some of the available studies use HPLC or UPLC instead of direct infusion, which due to the separation of isomers will lead to different results for the detected number of formulae, O/C etc.

It is therefore mostly only possible to compare general trends. In addition to the UPLC-MS measurements by Wang et al. (2018 & 2019), which are referenced on page 14, line 31, page 18, line 15 and page 19, line 20, we have now added references to results from Jiang et al. (2016) (p. 6, line 19).

12. Page 7, Line 14: You are right, here a reference would be helpful.

References were added and the sentence was changed to (p. 9, Line 11):

"Aromatic compounds are predominantly produced from anthropogenic sources such as traffic, industry and heating (Baek et al., 1991; Hamilton and Lewis, 2003) whereas aliphatic compounds can be of both anthropogenic and biogenic origin, the latter of which makes a larger contribution to organic particle mass in the summer (Gelencsér et al., 2007; Hu et al., 2017; Kleindienst et al., 2007)."

13. Page 7, Line 15: What does "high and low H/C particulate matter" mean?

The sentence has been reworded (p. 9, line 13):

"One type of source that will be present in both seasons and which contributes compounds of both high and low H/C to particulate matter is vehicle emissions, as these are usually a mix of low carbon number (<24) PAHs and single-ring aromatics with low H/C and alkenes and cyclic, branched and straight-chain alkanes with high H/C (Gentner et al., 2012, 2017; Huang et al., 2015; May et al., 2014; Worton et al., 2014) ."

14. Page 8, Lines 1-2: The SH showed high H/C ratio. The authors suggest that is due to a large proportion of biogenic organic aerosol from plant sources. It might be good here to give more evidences. Is there any assigned formula which could be used as markers for biomass burning or biogenic organic aerosol? For example, nitrophenols, nitrocatechols? They show higher number fraction and/or relative abundance in SH samples?

A reference for increased concentration of biogenic aliphatic compounds from plant waxes during summer in Beijing has been added (Feng, 2005).

Direct infusion high resolution MS does not allow for structural identification of compounds. Thus, we do not speculate about specific markers.

Nitrophenols and nitrocatechols have low H/C of typically below 1 and not high H/C near 2, as discussed here.

15. Page 8, Line 9: It might be better to give the ozone concentrations for the WH, WL, SH, SL samples.

We have now added the average ozone concentrations for the different samples, so that the text now reads (p. 10, line 8):

" The average ozone concentrations at the IAP site during sample collection were 13 ppb (WL), 6 ppb (WH), 39 ppb (SL) and 63 ppb (SH)."

16. Page 9: The VK plots show the aromaticity of CHO and CHON is quite different in winter and summer. Please speculate more in depth on the difference.
Differences in aromaticity between the winter and summer sample are discussed in depth in the following section, 3.2.

17. Page 10: Here the authors give some data on polyaromatic compounds. How about the single-ring aromatics? I think they contribute more to the aromatic compounds.
The relative dominance of monocyclic vs. polycyclic aromatics in the different samples is discussed in more depth on page 12, line 21 to 23 of the original manuscript. We have added some absolute numbers for comparison, so that this section now reads (p. 14, line 25):
"The SL sample is particularly low in aromatic formulae (132 aromatic formulae vs. 801 in total), especially regarding polycyclic aromatics (26 formulae). In contrast, there are a reasonable number of polycyclic (85) and monocyclic (159) aromatics in SH. The two winter samples both show high contributions of aromatic formulae (76% in WH vs. 77% in WL of all detected formulae in these samples). However, the WH sample is strongly dominated by polycyclic aromatics with 403 polycyclic vs. 260 monocyclic aromatic formulae, while monocyclic and polycyclic aromatics are present in nearly equal numbers in the WL sample (320 monocyclic and 297 polycyclic aromatic formulae)."

18. Page 12: The authors spend too many words to discuss the aromaticity equivalent (Xc). I suggest the authors focus on the new finding which cannot be deduced from the H/C and O/C ratios.
Excluding the introduction of the concept, the discussion of the aromaticity equivalent on page 14 is largely focused on the presence of monocyclic vs. polycyclic aromatics, which cannot be deduced from H/C and O/C ratios.

19. Page 13, Line 21: A reference regarding the heating source would be helpful.
We have added the following sentence (p. 15, line 29):
"This link between higher-ring number polyaromatics and residential heating is supported by studies showing a sharp increase in the concentration of higher-ring-number PAHs at the start of the heating season in Beijing (Zhang et al., 2017) and increased emissions of higher-ring-number PAHs for coal combustion compared to gasoline and diesel (Huang et al., 2014)."

20. Page 16, Line1: The authors state that sulfate concentrations are usually higher in summer than in winter in Beijing. Are you sure? Please give the concentrations of sulfate and nitrate in WH, WL, SH, and SL samples in a Table, for instance, in the Supplementary Information. And in Line 25, you state the lower sulfate concentrations in summer. It is significantly contradictory.
The average concentration of sulfur in fine PM in Beijing are indeed usually higher in summer, as shown in the four cited papers. This is often explained by the increase in photooxidation during summer. During our sampling campaign the average sulfate concentration was however slightly higher in winter (8.5 ug/m3) than in summer (6.9 ug/m3). This might be due to the fact that our campaign only covered early summer, whereas the very high sulfate levels found in the cited papers were all measured later in summer than ours. This is now explained in the paper to resolve this apparent contradiction (p. 18, line 7).
In addition, sulfate concentrations also increase during haze events, of which there were several very strong ones during our winter measurement campaign. For our study, samples were chosen at clean vs. polluted conditions. This explains the very high sulfur levels we see in winter since the extreme pollution events in winter lead to higher maximum sulfate concentrations. We agree that the sentence in line 25 is confusing in the original text. The wording has been changed to (p. 19, line 14):
"The sulphate data did not correlate as well for summer (Fig. 8b), which might be explained by the lower maximum concentrations of $SO_4^{2-}$ and the on average slightly lower $SO_4^{2-}$ concentrations during our summer campaign, where particle-phase formation reactions of S-containing organics might become less important."

We hope that this, in combination with the abovementioned explanation about the sulfate concentration during our sampling campaign vs. other summer measurements has clarified the issue.

21. Page16: It is a good idea to discuss the relationship between sulfate and nitrate with the number of CHOS and CHON compounds, but I recommend the authors to provide in-depth insights into this discussion. In addition to the secondary formation, Song et al. (EST 2019, 53, 13607-13617; 52, 2575-2585) reported that S-containing compounds account for 36% of the total number of compounds identified, making up the largest component in coal smoke, and N-containing compounds show high abundance in biomass burning. The primary sources of S- and N-containing compounds should also be considered.

The discussion of the formation of S- and N-containing compounds has been expanded (p. 18, line 12 & 28).

22. Page 16: The authors state that the particles in winter are sampled before they can undergo atmospheric ageing processes, for example reacting with OH radicals and ozone in Page 8, Lines 6-7. It seems inconsistent with the good positive correlation between sulfate and CHOS compounds.

As stated previously, the sulfate concentration depends not only on oxidation but also on $SO_2$ emissions, which are usually much higher during winter.

*Technical corrections:*

1. Page 5, Lines 27-28: What does "Fehler! Verweisquelle konnte nicht gefunden warden" mean? Mistype?

This has been corrected.

2. Page 5, Line 30: "off" should be "of".

This has been corrected

---

## Author Comment (AC2) · 20 Jul 2020

The authors thank the reviewer for taking the time to review this manuscript and for the constructive criticism.

This document includes authors' responses to anonymous referee #3 (RC2). Reviewer's comments are in black text while the authors' responses are in blue, with amended text quoted from the manuscript in quotation marks. Page numbers and lines refer to the revised version unless explicitly stated otherwise.

**Anonymous Referee #3 (RC2)**

*General comments:*

This study reports the analysis of organic aerosol composition in central Beijing during winter and summer months using orbitrap mass spectrometry coupled with a nanoelectrospray ionisation source. A main finding is that the number of S-containing organic species increased with inorganic sulfate concentration. This work provides new information on organic aerosol chemistry in northern China with a scope that fits well within ACP. But there are some issues on data analysis and interpretation, which may require major revisions to resolve. It is important that the authors discuss the limitations with negative ESI MS analysis, such as its low ionization efficiency towards, or inability to detect, certain types of C1 compounds.

Compounds with only one C atom usually have very high volatilities and thus are not expected to be present in the particle phase. We therefore do not discuss C1 compounds here.

A general comment on the ionisation efficiency of ESI was added on p. 6, line 9. A specific discussion of ionisation efficiency for N- and S-containing compounds has been added in the respective section (p. 17, line 10 & p. 19, line 27).

The calculation of aromaticity equivalent Xc is based on assumptions of elements' valences which may not always hold for atmospheric organics. The authors excluded S-containing species in their calculations of Xc, but nevertheless calculated the Xc for N-containing compounds, in which the valence of N can be either 3 or 5. In addition, the O valence in peroxides is 1 rather than 2 and it is known that organic peroxides can account for a significant fraction of the molecules. The authors should discuss how these issues affect their results and conclusions.

The relevant metric for the calculation of the aromaticity equivalent is not the valency but whether oxygen or sulfur atoms are present as π-bond structures in a particular compound, since the contribution of those structures is supposed to be removed when calculating the degree of unsaturation. For our calculations, we chose m=0.5, meaning that half of the oxygens in each functional group are present as π-bond structures, which is the case e.g. in carboxylic acids, which are likely dominant components in the sample. Peroxides, just like alcohols, only contribute σ-bond, so for them m=0. In the presence of many peroxide functions our assumption would lead to an underestimation of the degree of saturation. The same is true for organonitrates and peracids, since here only one in three O-atoms participates in a π-bond, rather than one in two as assumed with m=0. This is now discussed on page 14, line 11.

Additionally, some discussions on the technical aspects a bit vague and need clarification. See below for specific comments.

*Specific comments:*

This study only compares winter and summer compositional differences, thus is an overstatement to have a title of "Seasonal Differences . . ."

We have changed the title to "Differences in the Composition of Organic Aerosols between Winter and Summer in Beijing: a Study by Direct Infusion Ultrahigh Resolution Mass Spectrometry"

Page 2, how was sampling from Birminghan UK decided to be representative of a typical European urban background site?

The site is not a "representative background station", which would require a huge effort by comparing the aerosol composition at many different sites. A comparison with this site was chosen because it is one of the only urban background sites in Europe where the aerosol composition was characterized with the same method as used here assuring direct comparability with the current study. We have removed the word "typical" to avoid any misunderstandings.

Page 3, What's the sampling duration for the filters?

The sampling duration was 23 hours for each filter, as stated on page 3, line 9.

Page 3, Line 19, is the concentration corresponding to PM mass or OA mass? How was it known?

The concentration refers to PM mass, which was determined gravimetrically. Particle mass was used since OA mass was not known at the point of the mass spectrometric measurements. "total mass" has been changed to "total particle mass" to make this clearer.

Page 4: Line 1-2, what's the mass accuracy of the instrument?

The mass accuracy of the instrument was below 1.5 ppm. This has now been clarified in the methods section (p. 4, line 7):

"The mass accuracy of the instrument was below 1.5 ppm, which was regularly checked before the analysis."

Line 7 – ´ 9, this sentence is vague. More information is needed to clarify how this was done.

This section has been changed as follows (p. 4, line 15):

"The three repeat measurements of the blank filters for both high and low mass range were manually merged to yield four final blank files: low mass range winter, high mass range winter, low mass range summer and high mass range summer. Each of these merged blank files contains all masses from the three repeat measurements as separate data points."

I hope this makes it clearer. The consequence of this is that if the threshold for blank subtraction for a specific mass was met for any of the three repeat measurements, the peak was removed, i.e. a conservative approach.

Line 11, be specific about the threshold to remove signals and define how noises ´ are determined.

The noise levels differ for each sample and are based on fitting a normal distribution to a histogram of intensities. The sentence has been changed to (p. 4, line 19):

"In the first instance, all ions below the noise level, which was estimated based on fitting a normal distribution to a histogram of intensities, were removed from the spectrum." to clarify this. Further details about the procedure and the reasoning behind it can be found in the cited paper by Zielinski et al. (2018) cited on p. 4, line 19.

Line 12, "blank subtraction" usually means that all ions were ´ subjected to blank subtraction, but this sentence suggests only the ions less than 10 times of the blank level are removed. This is confusing.

In direct infusion ESI, signal intensities cannot be correlated directly to concentrations. Thus, we have a very conservative approach where we delete peaks from the mass spectrum when we identify a peak with the same exact mass in the blank with 10% or more intensity compared to the sample spectrum. Much more details are given in Zielinski et al. (2018).

Line 20 -21, the formula for DBE calculation has limitations due to the assumptions about element valences. This issue should be clearly stated and the implications on the reported results should be discussed.

The assumption of trivalence for nitrogen has now been explicitly stated (p. 4, line 32). The implications of the valence assumptions are now discussed in the results section (3.2 Aromatic compounds). In

short, the calculated DBE values represent a lower boundary on the DBE due to the assumptions about the valence state.

However, this lower limit is likely to be a better representation of the aromaticity of the compounds than one calculated with taking the higher valency into account since the additional double-bonds added by including the higher valencies are not contributing to aromaticity, ring formation or condensation.

Line 23 – 24, the sentence "If ´ . . ." is confusing. Please be specific.
The sentence has been changed to (p. 5, line 1):
"If there was no peak with a matching composition containing only the lighter isotope or if the intensity ratio of heavier-to-lighter isotope was greater than the natural isotopic abundance, the formula with the next larger mass error was used instead."
Further information about the process can be found in Zielinski et al. (2018).

Page 5: The Panagi et al. paper is not yet published and unavailable. It is not ´ C2 appropriate to cite it as a source of information used in this paper. Either provide the paper as supplementary or reiterate relevant key points.
This paper is now published and the proper citation has been added.

The meaning of "the residence time of the air masses (or the integrated concentration of theoretical air mass particles)" is not straightforward, needs clarification.
The model releases 1g/s of particles for 3 hours (so 3600 x 3 g = 10800g). Snap shots of the particle locations are taken every 15 minutes and then summed up to give the number of particles in each grid box during the past 72 hours so that for areas close to the station, if a particle doesn't move much it could be sighted 72 x 4 times! So, when we say relative residence time, the color scale is denoting whether there are lots or few particles, on a logarithmic scale.
We have changed this sentence accordingly in the caption, which we hope is clearer (p. 9, line 2):
"The colors denote the relative residence time (on a logarithmic scale) of the air masses in each 0.25 ° x 0.25 ° grid box (up to 100m from the surface) during the last 72 hours before arriving at the monitoring station (the model calculates the concentration of theoretical air mass particles in a grid box integrated over time)."

Line 22-24 seems unnecessary, consider to remove.
We have removed these sentences.

There are strange characters shown at Line 27 – 28.
This has been corrected.

Change "off" to "of" on Line 30
Done

Page 7. Line 14. Ref?
The appropriate references have now been added.

Page 8: Line 1-2, waxy biogenic organic aerosol components likely have high H/C, ´ but not all biogenic compounds have high H/C. It is more useful to define the cutoff value of "high H/C"
We agree that particles with biogenic sources do not necessarily have high H/C, we were here specifically referring to primary biogenic plant sources with high H/C. The sentence has been reworded as following to clarify this (p. 9, line 17):
" The SL H/C ratio is particularly high which may be due to a larger proportion of primary biogenic organic aerosol components from plant sources with a high H/C, such as plant waxes, and a smaller influence of industrial sources or vehicle emissions which is more pronounced in the high pollution sample."

line 14, what's the basis for claiming that compounds with H/C´ < 1 and O/C < 0.5 are aromatic? Citing a previous study here without proper context is not sufficient.

Non-alkylated (poly-)aromatic compounds have by definition an H/C < 1. The O/C ration is a softer limit but a mono-aromatic with > 3 oxygen groups would be a rather exotic structure. The same applies for PAHs.

Page 10, specify the "low" and "high" values use in describing elemental ratios and discussing chemical meanings.

"High" and "low" are used here to compare the numbers in the table in a relative and qualitative way. The chemical meanings of the elemental ratios are described on p. 10, line 3:

"While the overall H/C ratio shows whether the sample is more aromatic or aliphatic, the O/C ratio gives an indication of how strongly oxidised a sample is."

Page 12, 1st paragraph, N also has two valences, so what's the validity of calculating Xc for N-containing compounds using the given formula?

As mentioned previously, the important factor is the fraction of oxygen atoms in each compound participating in π-bonds. With our current assumption, this number is 0.5 which is true for the nitro group. For organonitrates, this number would be lower, leading to an underestimate of unsaturation using the current assumptions. This would also be the case for reduced nitrogen compounds such as e.g. amines, although the reviewer notes correctly that these are unlikely to be detected in negative mode ESI.

Page 16, line 33, what's the reasoning behind this sentence – "This suggests . . ."? Why does the correlation suggest how the compounds are formed? The authors appear to imply that the N-containing ions detected in this study are representative of " N-containing organics" in aerosol, but this is misleading as negative mode ESI-MS generally biases against reduced nitrogen compounds. Such issues should be articulated throughout the manuscript.

As mentioned earlier, any mass spectrometry method is biased against the compound classes which ionise best with the particular ionisation technique used. ESI negative mode ionisation is specifically sensitive towards oxidised N-compounds, not reduced functional groups such as amines, which ionise very efficiently in positive mode. This has now been clarified at the beginning of section 3.3 (p. 17, line 10).

We agree that the mentioned paragraph might be misleading and have therefore changed this to (p. 19, line 27):

"As stated earlier, the detection mode we used is biased towards oxidised N-containing organics such as organic nitrates, which are known to form predominantly in the gas phase and and nitro compounds, which can be of either primary or secondary origin, with a strong contribution of gas-phase oxidation in the second case. A correlation with particle-phase nitrate is therefore not expected."